# Foliar Infiltration of Virus-Derived Small Hairpin RNAs Triggers the RNAi Mechanism against the Cucumber Mosaic Virus

**DOI:** 10.3390/ijms23094938

**Published:** 2022-04-29

**Authors:** Bernardo Villegas-Estrada, Manuel Alejandro Sánchez, Arnubio Valencia-Jiménez

**Affiliations:** 1Departamento de Producción Agropecuaria, Universidad de Caldas, Manizales 170004, Caldas, Colombia; bernardo.villegas@ucaldas.edu.co; 2Servicio Nacional de Aprendizaje-SENA, Cúcuta 760004, Norte de Santander, Colombia; manushao87@gmail.com

**Keywords:** CMV, capsid protein, *Nicotiana benthamiana*, gene silencing, shRNA

## Abstract

Post-transcriptional gene silencing (PTGS) is an evolutionarily conserved plant defense mechanism against viruses. This paper aimed to evaluate a dsDNA construct (77 bp) as a template for in vitro production of virus-derived artificial small hairpin RNAs (shRNAs) and test for their potential to trigger the RNAi mechanism in *Nicotiana benthamiana* plants against CMV after their foliar infiltration. This approach allowed for the production of significant amounts of shRNAs (60-mers) quickly and easily. The gene silencing was confirmed using polymerase chain reaction (PCR), immunological-based assays, and real-time PCR (qPCR). The highest levels of gene silencing were recorded for mRNAs coding for replication protein (ORF1a), the viral suppressor of RNA silencing (ORF2b), and the capsid protein (ORF3b), with 98, 94, and 70% of total transcript silencing, respectively. This protocol provides an alternative to producing significant shRNAs that can effectively trigger the RNAi mechanism against CMV.

## 1. Introduction

Cucumber mosaic virus (CMV) was identified in 1916 as the causal agent of plant disease in cucumber and muskmelon in the United States [1]. CMV, widely spread worldwide [2], is a member of the genus *Cucumovirus*, belongs to the *Bromoviridae* family [3]. This virus is well known to generate severe damage and significant agronomic losses in many crops. It has been registered in more than 1300 species belonging to more than 100 botanical families, including monocots and dicots [4], making it one of the most studied plant viruses [5]. CMV virions appear as icosahedral particles of about 30 nm in diameter, composed of 180 subunits of a single capsid protein (CP) and 18% RNA [6]. The genome of this plant virus is composed of three single-stranded positive (+) sense RNAs that encode for five proteins [7]. Two protein subunits related to replicase and the movement protein are translated directly from the genomic RNAs. In contrast, CP and 2b proteins are translated from subgenomic RNAs 4 and 4A [8].

The CMV control usually relies on insecticides and integrated pest management techniques that target their specific biological vectors [9]. The frequent occurrence of pesticide-resistant insect populations and a general concern related to potential health risks for either farmers or consumers are associated with pesticides and their effects on beneficial insects [10]. These not wanted effects demonstrate the need to design and evaluate alternative protection methods against plant viruses attacks.

RNA interference (RNAi) technology has produced promising results by down-regulating the expression of viral genes and endogenous plant genes [11]. Post-transcriptional gene silencing (PTGS) in plants is well known as an evolutionarily conserved defense mechanism against pathogens, especially against viruses [12]. PTGS can be triggered either by self-complementary hairpin structures produced during the virus replication or by double-stranded RNA (dsRNA) containing the homologous sequence to that of the target gene [13]. The plant machinery processes silencing—inducing molecules into single-stranded small-interfering RNA (siRNA) molecules that are then incorporated into an RNA-induced silencing complex (RISC) that subsequently degrades the homologous RNA in a sequence-specific manner [12]. As new research on gene silencing continues to appear, it is even more evident and proven that hairpin RNA (hpRNA)-induced silencing represents a valuable and powerful biotechnological tool to develop resistance against plant viruses [14,15]. Recently, several studies provided evidence that exogenous application of dsRNAs on plant tissues induced RNAi-mediated silencing of the targeted genes [16,17,18,19]. This paper presents the design, synthesis, and gene silencing effect triggered by artificial small hairpins (shRNAs), targeting four CMV genes that play roles in genome replication (ORF1a and ORF2a), encapsidation (ORF3b), and counter defense (ORF2b). The size and how the shRNAs were synthesized, in addition to their infiltration as naked and non-conjugated forms used in this approach, proves that this is a valuable and alternative strategy to evaluate the potential of shRNAs in triggering the RNA silencing mechanism against plant viruses.

## 2. Results

### 2.1. Synthesis, Quality, and Yield of ShRNAs

Equimolar amounts of the reverse complementary DNA oligos designed for the four open reading frames present in the positive (+) genomic RNA segments of CMV, containing a complementary 25-base pairs (bp) long region, the T7 promoter region, and a seven-nucleotide loop were annealed using a process of heating and cooling. The designed final DNA construct (77 bp) proved to be suitable for synthesizing shRNAs, yielding hairpin molecules of about 60 mers in length (Figure 1). When using just 8 µg of template DNA, about 300 µg shRNA (with a concentration approx. of 10 µg/µL) were obtained in a 30 µL final reaction volume. Synthesis results, using this approach, show that it is possible to obtain 10 mg of artificial shRNA using just 130 µg of each DNA oligo before the hybridization protocol. This total amount of synthesized shRNA was then used as a biotechnological tool to trigger the plant RNA silencing mechanism against the cucumber mosaic virus.

Each synthesized shRNA was observed as a clear band around the expected size (60-mers long) in the agarose gel electrophoresis. The ORF2b translated respectively into viral suppressor of RNA silencing was much better synthesized in vitro (Figure 2). The synthesized shRNAs were not degraded when they were treated with either DNase or RNase A, revealing a resistant hairpin-type RNA secondary structure (Figure 2).

### 2.2. Primer Efficiency and Specificity

Primer efficiency (E%) values ranged between 92.76 and 110%, and correlation coefficients (R2) had values above 0.955 (Table 1). RT-qPCR products were confirmed by the presence of a single peak in melting curve analyses and visualized in a 1.5% agarose gel.

### 2.3. Effect of Virus-Derived Artificial ShRNAs on CMV Infection

Control and shRNAs-treated *N. benthamiana* plants were initially infiltrated with a mix containing each virus-derived artificial shRNAs shRNA (200 ng/µL of each specific shRNA) and were inoculated with CMV a day later to assess the capability of these small hairpin RNAs to inhibit CMV infection. This protocol resulted in 100% of negative control and shRNA-treated plants being free of symptoms and appearing resistant to CMV after 15 dpi (Figure 3a). Mild chlorosis was observed in shRNA-treated plants after 15 dpi (Figure 3c). On the other hand, 66% of the positive control (H_2_O/CMV) seedlings showed chlorosis and leaf curling at 10 dpi, and 100% of these seedlings showed similar symptoms just five days later (Figure 3b). Overall, symptoms detected on leaves after infiltration and inoculation treatments were consistent with ImmunoStrip^®^ results at 7 dpi (Figure 3a–c).

### 2.4. Immunological-Based Assays for Specific Detection of CMV

To estimate virus accumulation in treated leaves, besides the visual inspection of foliar symptoms, both dot-blot and ELISA, focused on CMV-capsid protein (CP) detection, were performed in foliar tissue samples from treated leaves collected at 10 and 2–3 dpi, respectively. Interestingly, plants challenged with CMV 1 day after being infiltrated with shRNA showed a delay of two weeks in the appearance of viral symptoms when compared to the positive control (H_2_O/CMV-treated plants). Besides, these plants also showed a lower capsid protein expression (CP) than positive control plants, as shown in the dot-blot immunological assay performed 10 days after CMV inoculation (Figure 4).

ELISA results indicate that at 2 dpi, the CMV-antigen (CP) accumulated higher in positive control plants (1.82 OD/mg of protein). This immunological assay failed to detect CMV capsid protein (CP) in shRNA/CMV-treated plants simultaneously (Figure 5a). However, ELISA immunoassay was able to detect 0.35 OD/mg of protein in shRNA/CMV-treated plants at three dpi when compared to its control treatment (1.32 OD/mg of protein) (Figure 5b). It is essential to point out that all the asymptomatic plants were ELISA negative under our assay conditions. In contrast, the *N. benthamiana* plants with apparent symptoms of CMV infection were ELISA positive (data not shown).

### 2.5. RT-PCR and RT-QPCR

shRNAs containing 25 nt of uninterrupted RNA duplex were synthesized to four target ORFs present in the three single-stranded positive (+) genomic RNA segments of CMV to determine if small hairpins (<30 nt) can trigger the RNA silencing mechanism against CMV. Artificial small hairpin effectiveness in silencing CMV gene expression was analyzed at the transcript level by monitoring foliar mRNA accumulation of target genes by regular RT-PCR and RT-qPCR. RT-PCR specific primers allowed the amplification of gene fragments related to target ORFs in samples from the positive control treatment (H_2_O/CMV-treated plants) (Figure 3d). Besides, a lower amount of RT-PCR product was generated when RNA samples from silencing treatments were used as template. However, it was impossible to obtain RT-PCR amplification from negative control samples (data not shown). RT-qPCR experiments showed a lower CMV gene expression of all target genes in *N. benthamiana* seedlings treated with shRNAs when compared to control samples (Figure 6). However, it is important to mention that high levels of gene silencing were recorded for mRNAs coding for replication protein (ORF1a), the viral suppressor of RNA silencing (ORF2b), and the capsid protein (ORF3b), with 98, 94, and 70% of total transcript silencing, respectively (Figure 6a,c,d). A significant reduction of transcript expression was also recorded for the replication gene (ORF2a), with 50% of complete transcript silencing (Figure 6b). Together, these results show that relative expression levels of the evaluated target genes correlate with ImmunoStrip^®^ tests results and the intensity of the observed symptoms in challenged plants.

## 3. Discussion

During the last years, many research studies have been focused on controlling the multiplication and the spreading of the cucumber mosaic virus, especially considering its great potential to generate substantial damage to agronomically important crops, leading to significant economic losses. The most durable and traditional method that can be used to introduce resistance genes into host plants is through classical breeding, which is usually very time-consuming. Progress on the implementation (manipulation, harnessing, etc.) of the post-transcriptional gene silencing mechanism shows that plant resistance to infection caused by a virus can be obtained using this way. RNA silencing is considered an evolutionarily conserved mechanism that plays a crucial role in controlling gene regulation in eukaryotes and a natural defense mechanism against viruses and transposable elements [20]. Plant antiviral defense is proposed to be a biological reason for the evolutionary development of the RNAi mechanism in plants [21]. Through this natural mechanism, the plant RNAi machinery can successfully limit viral infection, thus controlling the final accumulation of viral RNA by converting viral dsRNAs into small interfering RNAs (siRNAs). It is well known that hairpin RNA constructs sharing homology to viruses and viroids have been evaluated for their potential to confer virus plant resistance [22]. This study used the RNAi mechanism to knock down the expression of four genes present in the three single-stranded positives (+) genomic RNA segments of CMV, using shRNAs synthesized in a rapid and straightforward protocol. In this approach, 4 µg of each of the two reverse complementary DNA oligos targeting four of the five ORFs present throughout genomic single-stranded (+)-RNA segments of CMV, containing a complementary region of 25 base pairs (bp) long, a seven-nucleotide loop, and the T7 promoter region, were successfully annealed. The final dsDNA construct (77 bp) was finally used as the DNA template for in vitro transcription protocol to produce about 300 µg of shRNAs (60-mers). Conventional production of small interfering RNA (siRNA) usually relies on recombinant DNA technology by constructing DNA-based vectors that can express hairpins, targeting specific viral genes [23,24,25]. This procedure usually requires several rounds of restriction and ligation reactions, making it tedious and time-consuming. An alternative synthesis method was successfully applied to generate significant amounts of shRNAs in a fast way, as revealed by 1.5% agarose gel electrophoresis, without the need of considering the traditional methodological approaches associated with recombinant DNA technology. Multiple studies have reported down-regulation in gene expression of specific viral genes via RNA interference (RNAi) after external application of RNA molecules [16,18,26,27,28], leading to a substantial induction of plant resistance against this kind of pathogen. In all those studies, reducing mRNA levels of target genes was possible due to the exogenous application of double-stranded RNA molecules. However, recent studies have reported plant viral resistance after external application of hpRNAs, targeting essential genes of the plant pathogen [29,30]. In those studies, the induction of gene silencing was achieved using viral-based hpRNAs from crude nucleic acid extracts produced by hpRNA-expressing bacterial strains. This study performed an in vitro method that uses a synthetic DNA construct, containing inverted repeats of CMV-specific sequences as a helpful template for producing virus-derived artificial shRNAs. This method does not require methodological approaches associated with recombinant DNA technology, such as constructing DNA-based vectors or regular PCR, except when large amounts of hpRNAs are needed. Results also show that the structure of the resulting transcripts favors the rapid formation of stable shRNA-type molecules that can trigger positive RNAi silencing responses in *N. benthamiana* plants, as demonstrated by immunological-based assays, RT-PCR, and RT-qPCR. DNA-based nanostructures conjugated to sRNAs were recently used to facilitate the delivery of 21 nt Green Fluorescent Protein (GFP) sRNAs into infiltrated *N. benthamiana* leaves [31]. This research work showed that sRNAs tethered to 3D nanostructures exhibited mRNA degradation of the GFP. Interestingly, results show that it is possible to maintain the plant protection effect for at least 15 dpi using just a unique 25 µg dose of the synthetic virus-derived artificial shRNA. In this study, the direct infiltration of naked and non-conjugated shRNAs has been shown to effectively trigger RNA silencing mechanisms against CMV, placing the *N. benthamiana* plants in an advantageous position to fight against this kind of pathogen. It is essential to point out that, although carrier-like compounds greatly facilitate RNA delivery into plant cells, they are also expensive and challenging to synthesize. The findings show that our approach can successfully be used to produce interesting amounts of shRNA that can effectively trigger virus gene silencing in preliminary plant RNAi-based experiments, without being immersed in expensive and time-consuming protocols that rely on the DNA-based vectors and in subsequent cell-based transformation. If necessary, synthesis of the dsDNA template and the synthesis of shRNAs can be escalated up to produce higher amounts of specific hairpins. Further studies such as the dose of shRNA, sequence length, and persistence on the leaves should be performed before making practical the use of shRNAs for field application.

We believe that direct infiltration of naked and non-conjugated shRNAs produced using this approach could be used to evaluate their potential to knock down the gene function of several viral and pathogen-related genes. We strongly believe that our laboratory protocol makes specific shRNAs with high quality, significant yield, and low cost.

## 4. Materials and Methods

### 4.1. Design and Synthesis of ShRNAs

Two reverse complementary DNA oligos for each of four open reading frames (ORF) present in the three single-stranded positive (+) genomic RNA segments of CMV (NCBI accessions: NC_002034.1; NC_002035.1, and NC_001440.1) were designed to contain a complementary region 25 base pairs (bp) long, a seven-nucleotide loop (5′-TCAAGAG-3′), and the T7 promoter region (TAATACGACTCACTATAGGG) at the 5′ end (Table 2). Three thymine nucleotides (TTT) were added at the 3′ ends of each oligo to stabilize synthetic shRNA better, as described by [32]. The secondary hairpin structure was confirmed using the RNA Secondary Structure platform (https://rna.urmc.rochester.edu/RNAstructureWeb/Servers/Predict1/Predict1.html (accessed on 1 March 2021)) and the designed oligos sent for synthesis to Invitrogen (Thermo Fisher Scientific, Waltham, MA, USA). The two single-stranded DNA (ssDNA) oligonucleotides (100 µM) with complementary sequences were subject to an annealing protocol. Briefly, each oligonucleotide was dissolved in 10 mM Tris, pH 7.5 containing 50 mM NaCl and 1 mM EDTA, mixed in a PCR tube at equal equimolar concentrations, and incubated at 95 °C for 5 min. Microtubes were allowed to cool down to room temperature to facilitate oligos hybridization slowly. The resulting duplex oligonucleotide was stored at −20 °C until used as templates for in vitro synthesis of shRNAs using the HiScribe^TM^ T7 High Yield RNA Synthesis Kit (Cat. E2040S, New England Biolabs, Ipswich, MA, USA) (Figure 1).

Following the manufacturer’s instructions, the synthesized shRNAs were purified using the RNeasy Mini Kit (Cat. 74104, Qiagen, Valencia, CA, USA) and then quantified using a NanoDrop ND-1000 spectrophotometer (Thermo Fisher Scientific, Waltham, MA, USA). To determine the stability of the synthesized shRNAs, the product was digested with DNAse I (Cat# M0303, New England Biolabs, Ipswich, MA, USA) and RNase (Cat# M0245S, New England Biolabs, Ipswich, MA, USA) following the manufacturer’s instructions. After digestion, their purity and integrity were evaluated by 1.5% agarose gel electrophoresis.

### 4.2. Plants, Virus Maintenance, and ShRNA Infiltration

*Nicotiana benthamiana* plants were used as the propagative CMV host for all experiments. Seeds were initially submerged in water for three days under constant light conditions as a pre-germination stage. After germination, seedlings were kept in sterile sand and soil (1:2) in 10 cm wide pots at 25 °C, 65–70% humidity, and 18 h/6 h light/dark cycle. The CMV viral isolate was provided by Dr. Wilmer Cuellar, Virology Unit of the International Center for Tropical Agriculture-CIAT (Palmira, Colombia) as freeze-dried leaf samples and then propagated in *N. benthamiana*. Negative control (H_2_O/H_2_O-treated plants), positive control (H_2_O/CMV-treated plants), and shRNA-treated plants were subjected to mechanical transmission of CMV virus. In this case, plant leaves (two leaves/plant) with a diameter between 1.5–2.0 cm were rubbed with 50 μL of fresh sap, which was prepared using 100 mg of CMV-infected leaf samples macerated at 4 °C with 1.5 mL of sterile distilled water. Twenty-four hours before challenging, the shRNA molecules were applied together, via infiltration, using a mix of 100 µL containing each shRNA (200 ng/µL of each specific shRNA) to *N. benthamiana* leaves. Control and shRNA-treated leaves were harvested 24 h after CMV inoculation and flash-frozen in liquid nitrogen, and then were stored at −80 °C until their use in RNA extraction for antibody-based assays and qPCR analysis. All plants were grown in the conditions mentioned above and showed symptoms monitored daily until 25 days post inoculation (dpi). Forty-five plants were used in two bio-assays for all analyses (Appendix A).

### 4.3. Protein Extraction and Quantitation

Total protein was extracted from control and shRNA-treated *N. benthamiana* foliar tissue challenged with CMV using the Minute^TM^ commercial kit Total Protein Extraction Kit for Plant Tissues (Cat. SD-008/SN-009, Invent Biotechnologies, Plymouth, MN, USA) and following the instructions given by the manufacturer. The Pierce ™ BCA Protein Assay Kit (Cat. 23227, Thermo Fisher Scientific, Waltham, MA, USA) was used for quantitation of total soluble protein. Bovine serum albumin (BSA) was used as a control protein. A calibration curve was generated using BSA concentrations of 25, 50, 200, 400, 800, and 1600 μg/mL. Protein in each standard solution was calculated by triplicate.

### 4.4. Immunological-Based Assays for Specific Detection of CMV

In addition to common plant symptoms, the effectiveness of the CMV inoculation was confirmed by using the ImmunoStrip^®^ CMV-specific kit (Agdia Inc., Elkhart, IN, USA) following the manufacturer’s protocol. For all gene silencing evaluations in treated leaves (in situ), we performed the dot-blot technique and enzyme-linked immunosorbent assay (ELISA) (Agdia Inc., Elkhart, IN, USA). For dot-blot analyses, aliquots of foliar tissue samples containing an equal amount of protein were manually spotted on the nitrocellulose membrane (Cat. #162-0116, Bio-Rad, Hercules, CA, USA) previously wetted with 20% methanol for 30 min. As the samples dried, the nitrocellulose membrane was blocked for 2 h at room temperature using 10 mL of a nonfat dry milk solution (4%) (Cat. 1706404, Bio-Rad, Hercules, CA, USA) diluted in phosphate-buffered saline (PBS) (Cat. 1610780, Bio-Rad, Hercules, CA, USA). The nitrocellulose membrane was incubated for 2 h with the CMV primary capture antibody (Cat. 44501/0500, Agdia Inc., Elkhart, IN, USA), washed three times for 10 min with PBS solution (0.5% Tween, pH 7.4), and incubated for 1 h with the secondary anti-rabbit IgG-Peroxidase antibody (Cat. 6154, Millipore Sigma, St. Louis, MO, USA). Secondary antibody was diluted 1:5000 in 4% nonfat milk/PBS. After regular washing steps, the reactive proteins on the nitrocellulose membrane were incubated at room temperature for 30 min with the horseradish peroxidase substrate (Cat. 170-8235, Bio-Rad, Hercules, CA, USA), following the manufacturer’s protocol. The final reaction was stopped washing the membrane with pure water and letting it dry out at room temperature.

Protein samples from control and shRNA-treated foliar tissue were also evaluated by ELISA (SRA 44501/0500, Agdia Inc., Elkhart, IN, USA), using polyclonal antibodies to capture the CP particles of the CMV and monoclonal antibodies conjugated with alkaline phosphatase for final detection. The absorbance at 405 nm was measured after 45 min of incubation with the substrate using an ELISA SmartReader 96 microplate reader (Accuris instruments, Edison, NJ, USA).

The dot-blot test included three independent biological replicates, while the ELISA test included six (Appendix A). In all cases we included three technical replicates.

### 4.5. RNA Isolation and CDNA Synthesis

Total RNA was extracted from 0.1 g of *N. benthamiana* leaf tissue using the RNeasy Mini Kit (Cat. 74104, Qiagen, Valencia, CA, USA) according to the manufacturer’s protocol, and then stored at −80 °C until use. The RNA integrity was assessed through 1.5% (*m*/*v*) agarose gel electrophoresis, and its concentration was estimated on a NanoDrop 1000 (Thermo Fisher Scientific, Waltham, MA, USA). cDNAs were synthesized from 1000 ng of total RNA with specific primers using the Maxima H Minus First Strand cDNA Synthesis Kit (Cat. K1651, Thermo Fisher Scientific, Waltham, MA, USA). After synthesis, cDNA samples were diluted (1:5) before being used in either PCR or qPCR.

### 4.6. PCR, Quantitative Real-Time PCR (qPCR) and Primer Efficiency Test

Primers were designed using the OligoPerfect Primer Designer (Thermo Fisher Scientific, Waltham, MA, USA) and validated by analysis of their PCR amplification efficiencies (E%) and correlation coefficients (R2). The length for all qPCR products was kept between 120–190 bp. Efficiency (E%) of all primers were calculated according to the equation: E = (10[−1/slope] − 1) × 100. Relative standard curves were generated with serial dilutions of positive control cDNA (1/3, 1/9, 1/27, and 1/81). The qPCR protocol was performed with SsoAdvanced Universal SYBR^®^ Green Supermix (Cat. 1725271, Bio-Rad, Hercules, CA, USA) on a StepOne Real-Time PCR System (Applied Biosystems Inc., Foster City, CA, USA). The master mix (30 µL) contained 15 µL SYBR green, 6 µL diluted (1:5) cDNA, 1.5 µL (10 µM) primers (F + R), and 7.5 µL of nuclease-free water. The qPCR was run at 95 °C for 3 min (holding stage), followed by 95 °C for 30 s, then 60 °C for 30 s (cycling stage). A melting curve was also generated to confirm the presence of a single amplification peak and to rule out the possibility of primer dimers’ formation. F-Box gene amplification was used as the reference gene. Three independent biological (Appendix A) and technical replicates were included in all cases in each qPCR run.

### 4.7. Statistical Analysis

Data from ELISA and RT-qPCR assays were analyzed using non-parametric Kruskal–Wallis analyses, with *p* < 0.05 as the significance level; while the comparative 2^−ΔΔCT^ method [33] was used to calculate the relative expression level of the target genes as compared to control treatment in RT-qPCR assays.

All statistical analyses were performed with the statistical software Infostat (ver. 2008).

## 5. Conclusions

Virus-derived artificial small shRNAs can trigger the RNAi mechanism in *N. benthamiana* plants when applied using foliar infiltration.

shRNA-triggered events can lead to detectable levels of gene silencing, thus negatively affecting the gene expression of target genes.

## Figures and Tables

**Figure 1 ijms-23-04938-f001:**
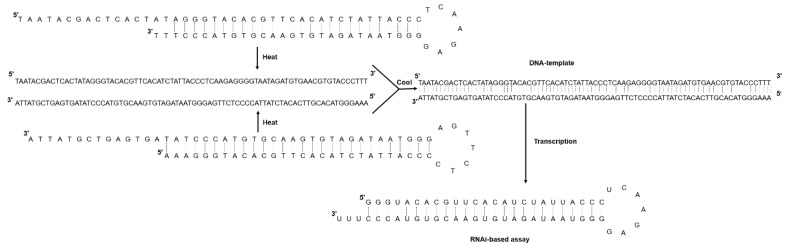
Schematic diagram representing the annealing reaction of two single-stranded DNA oligonucleotides with complementary sequences to form a shRNA. Heat at 95 °C for 5 min disrupts any secondary structure, and slow cooling facilitates hybridization between the complementary oligo sequences.

**Figure 2 ijms-23-04938-f002:**
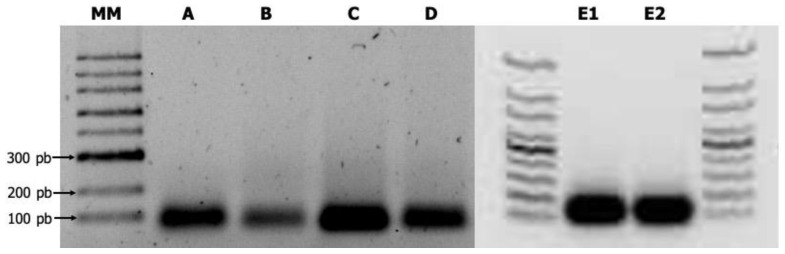
Produced virus-derived artificial shRNAs of CMV as revealed by 1.5% agarose gel electrophoresis. MM. 100-bp DNA ladder (New England Biolabs, Ipswich, MA, USA). A. shRNA-1; B. shRNA-2a; C. shRNA-2b; D. shRNA-3b; E1. shRNA-1 digested with DNase. E2. shRNA-1 digested with RNase A.

**Figure 3 ijms-23-04938-f003:**
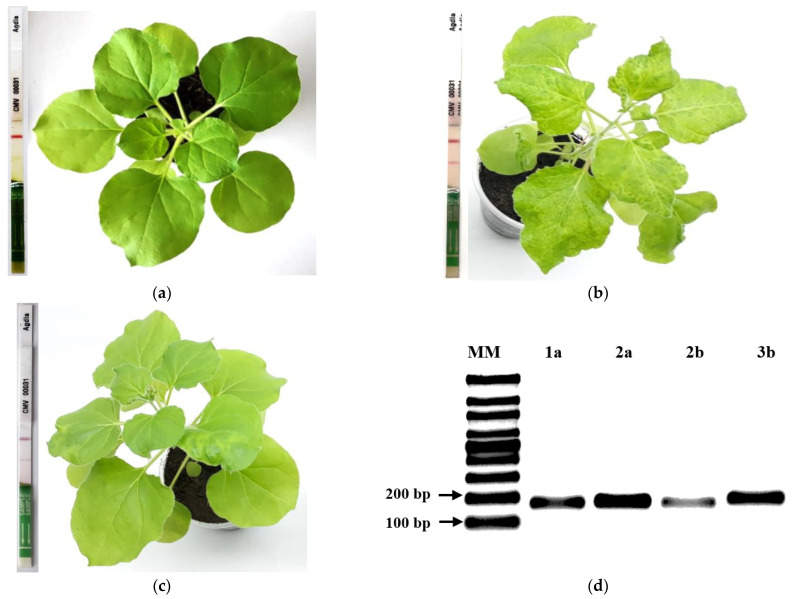
Effects on disease incidence of CMV on *N. benthamiana* plants at 15 dpi of virus-derived artificial shRNA, ImmunoStrips^®^ test results by treatment at 7 dpi and PCR results of positive control for four CMV genes. (**a**) Negative control showing just one line in test strip and plant without symptoms (H_2_O/H_2_O-treated plants); (**b**) Positive control showing two lines in test strip and plant with typical viral mosaic symptoms (H_2_O/CMV-treated plants); (**c**) shRNA/CMV-treated plants showing one strong and another mild line in test strip, and a plant with mild chlorosis. CMVBars scale = 1 cm. d. RT-PCR amplification of CMV replication (1a, 2a), suppressor of silencing (2b) and capsid protein (3b) (**d**) PCR analysis of four CMV genes (H_2_O/CMV-treated plants).

**Figure 4 ijms-23-04938-f004:**
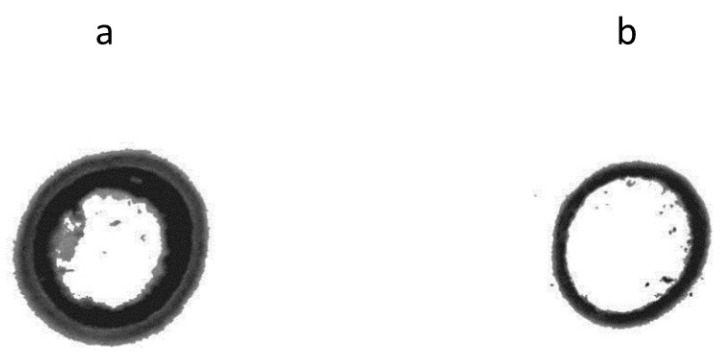
Dot blot for CMV-capsid protein (CP) detection on treated leaves at 10 dpi. (**a**) Positive control (H_2_O/CMV-treated plants); (**b**) shRNA/CMV-treated plants (this image represents one of three replicates used in this assay).

**Figure 5 ijms-23-04938-f005:**
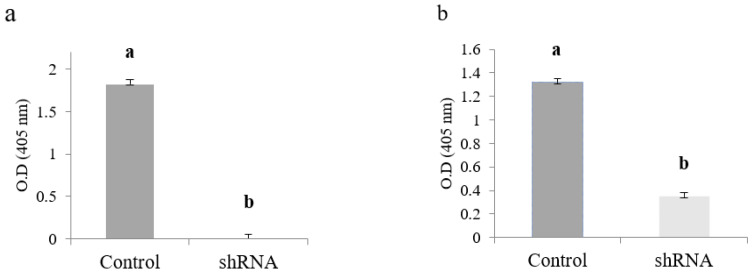
Effect of the knockdown of the CMV-capsid protein (CP) gene on the protein expression in *N. benthamiana* foliar tissue measured by direct ELISA. (**a**) Evaluation at 2 dpi; (**b**) Evaluation at 3 dpi. Values are shown as the mean and standard errors (±SE) of three biological replicates, each with three technical replicates. Different letters (a, b) represent significant differences at *p*-value < 0.05 according to the Kruskal–Wallis test.

**Figure 6 ijms-23-04938-f006:**
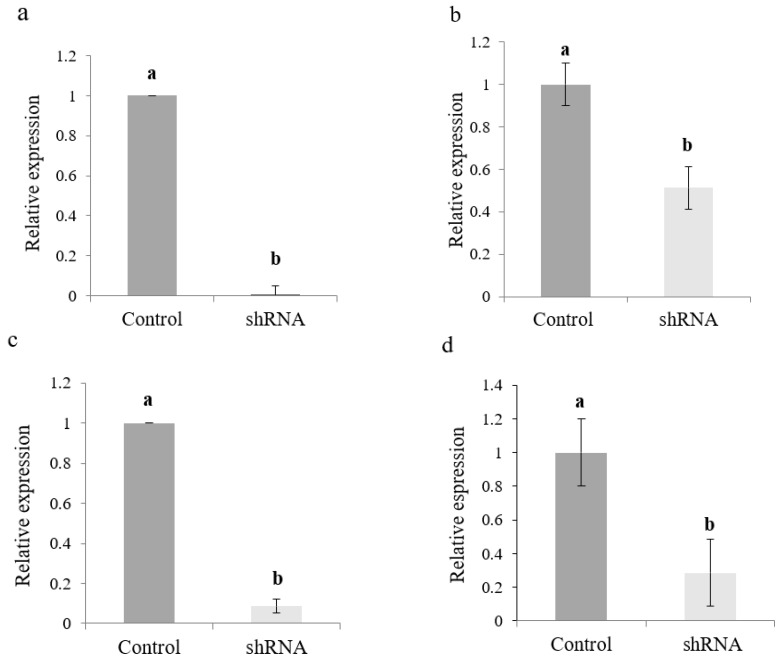
Relative transcript levels of four CMV genes evaluated in *N. benthamiana* foliar tissue by RT-qPCR at 3 days of virus-derived artificial shRNA exposure. (**a**) Gene 1a; (**b**) Gene 2a; (**c**) Gene 2b; (**d**) Gene 3b. Values are shown as the mean and standard errors (±SE) of three biological replicates, each with three technical replicates. Different letters (a, b) represent significant differences at *p*-value < 0.05, according to the Kruskal–Wallis test.

**Table 1 ijms-23-04938-t001:** Primer sequences of target and reference genes used in RT-qPCR analysis.

Genes	Sequences (5′-3′)	Amplicon Size (bp)	E (%)	R^2^
Replication (1a)	F: GCGTTATCCACGCTGGTATT R: AAATCCGCACTGTTTTCCAC	165	110.00	0.996
Replication (2a)	F: TGGATGTCAGCGAGAGTGTC R: ATACGCATGGGTTTGACCAT	172	103.36	0.981
Suppressor of silencing (2b)	F: CAAAAGTCCCAGCGAGAGAG R: GGCGAACCAATCTGTATCGT	190	94.85	0.955
Capsid (3b)	F: AACCAGTGCTGGTCGTAACC R: GCGTTCACTCCCTACAAAGG	172	92.92	0.995
*F-Box* (Reference gene)	F: GGCACTCACAAACGTCTATTTC R: ACCTGGGAGGCATCCTGCTTAT	127	92.76	0.994

**Table 2 ijms-23-04938-t002:** Reverse complementary DNA oligos (77 bp) designed to synthesize virus-derived artificial shRNA of CMV.

Oligo	Strand	Sequences (5′3′)	Gene Function
shRNA1a	Sense	TAATACGACTCACTATAGGGTATTGTTTATTCTGTCGGTTATtcaagagATAACCGACAGAATAAACAATACCCTTT	Replication (ORF1a/RNA1)
	Antisense	AAAGGGTATTGTTTATTCTGTCGGTTATctcttgaATAACCGACAGAATAAACAATACCCTATAGTGAGTCGTATTA	
shRNA2a	Sense	TAATACGACTCACTATAGGGTCCATTTTTGGTACCCGTGAAGtcaagagCTTCACGGGTACCAAAAATGGACCCTTT	Replication (ORF2a/RNA2)
	Antisense	AAAGGGTCCATTTTTGGTACCGTGAAGctcttgaCTTCACGGGGTACCAAAAATGGACCCTATAGTGAGTCGTATTA	
shRNA2b	Sense	TAATACGACTCACTATAGGGTCATGCCGCCATGTGAACGTGGtcaagagCCACGTTCACATGGCGGCATGACCCTTT	Silencing suppressor (ORF2b/RNA2)
	Antisense	AAAGGGTCATGCCGCCATGTGAACGTGGctcttgaCCACGTTCACATGGCGGCATGACCCTATAGTGAGTCGTATTA	
shRNA3b	Sense	TAATACGACTCACTATAGGGTACACGTTCACATCTATTACCCtcaagagGGGTAATAGATGTGAACGTGTACCCTTT	Capsid (ORF3b/RNA3)
	Antisense	AAAGGGTACACGTTCACATCTATTACCCctcttgaGGGTAATAGATGTGAACGTGTACCCTATAGTGAGTCGTATTA	

Each oligo contains the T7 promoter region (bold and underlined nucleotides) at the 5′ end, a complementary region of 25 base pairs (bp) long, separated by a seven-nucleotide loop (lowercase nucleotides), and three thymine nucleotides at the 3′ end (underlined nucleotides).

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
