# Peer review of "Foliar Infiltration of Virus-Derived Small Hairpin RNAs Triggers the RNAi Mechanism against the Cucumber Mosaic Virus"

_ijms, 2022, doi:10.3390/ijms23094938_

Round 1

Reviewer 1 Report

The paper by Villegas-Estrada et al. describes the in vitro synthesis of short hairpin RNA (shRNA) molecules that were used to induce RNA silencing aiming to control the infection by cucumber mosaic virus. Four shRNA molecules were generated which target different targets in the viral genome. The use of these molecules induced a delay in symptom appearance, lower viral genome loads, and a reduction in the accumulation of viral coat protein.

Altogether, the manuscript covers relevant information, particularly concerning the generation of shRNA, but several points require attention before its final evaluation.

The main points are as follows:

  • It is not clear whether the four generated shRNA molecules were applied together or individually. If this were the case, add the missing data.
  • Statistic assessments of some experiments seem not to be accurate (see below).
  • Figures show a limited representation of samples. I mean, only one plant per treatment (Figure 3), only one dot (Figure 4). I strongly suggest adding more pictures providing better support for the results.
  • Some results are not described, e.g. ImmunoStrips® test shown in figure 3.
  • Section 2.2. Primer efficiency and specificity. It is not clear the aim of this section. I believe that Table 2 (which should be Table 1) describes the efficiency of primers used in the qPCR assays, doesn’t it? In that case, this information could be shown as a Supplementary Table.

Other aspects to be improved:

Line 25: Bromoviridae in italics

Line 25: “… , widely spread worldwide” Are you qualifying the genus, the family, CMV, or everything? Clarify it.

Line 34: Move this sentence to the end of the former paragraph.

Line 56: “shRNAs” short hairpin RNA? Introduce it.

Line 74:  against cucumber mosaic virus. Don’t use a capital letter in this case.

Line 80:  “The ORF2b translated respectively into viral suppressor of RNA silencing was much better expressed” ??? How many replicates?

RNAse reaction conditions????

Line 94: Table 1 or 2???

Figure 2 E1 and E2. Indicate which of the shRNA was used in these assays. 

Figure 4: Could you show a larger number of samples per treatment? Besides, please include dots corresponding with non-inoculated plants treated with the shRNA solution and healthy-non-treated plants.

Line 99: “This protocol resulted in 100%  negative control and shRNAs-treated plants being free of symptoms and appearing resistant to CMV after 15 dpi (Figure 3a). Mild chlorosis was observed in shRNA-treated plants between 15 and 25 dpi (Figure 3c).” These two phrases convey potentially contradictory information.  After 15 dpi and between 15 and 25 dpi. What was the ratio of shRNA-treated plants with mild chlorosis after 15 dpi?

Line 102: “Overall, symptoms detected on leaves after infiltration and inoculation treatments were consistent with ImmunoStrip® results (Figure 3a-c).” What are the results with ImmunoStrip” Describe the results in the text.

Line 122: “ELISA results indicated that at…”

Line 125: Remove “However”.

Line 125: “ELISA immunoassay was able to detect 0.35 OD/mg of protein…..”

Figure 5: Legend. “Different letters represent significant differences at p-value < 0.05.” Letters were not included. By the way, the bars indicating the standard errors seem to be of the same length. It is correct???? Please, verify the original data.

Line 136: RT- PCR and RT-qPCR

Results: The headings of this section seem to describe an M&M section e.g. “2.1 Synthesis, quality, and yield of shRNAs, 2.5. PCR and qPCR. Please, select phrases that briefly describe the results obtained in each subsection to be used as subsection headings.

Line 144. “a lower amount of PCR product when using cDNA from silenced leaves.” Where these results were shown? Besides, I think that plant leaves were not silenced. Please write this phrase.

Figure 6. As previously observed in Figure 5, bars of standard errors in the treatments represented in figures 6 b and c seem to be of the same length. It can happen, but indeed, it is very uncommon. Please, verify the original data and the derived calculation.

  • Legend: “a. Gene 1a. b. Gene 2a. c. gene 2b and d. gene 3b.” G or g, I think G rather than G. First in the phrase.
  • Could you add letters as a result of the statistical analysis?

Line 186: Progress on the implementation (manipulation, harnessing, etc.) post-transcriptional gene silencing mechanism …

Line 238: “….at the 5' end. (Table 1).” Remove dot after end. I think there is something wrong with the numbering of tables. This should be Table 2.

Line 262: “Nicotiana benthamiana plants were”

Line 285: “The Pierce ™ BCA Protein Assay Kit (Cat. 23227, 284 Thermo Scientific, Rockford, USA) was used for CP quantitation.” I can not understand how this assay specifically detected CP, rather than total protein from the plants. Explain.

Author Response

Main points:

Point 1: It is not clear whether the four generated shRNA molecules were applied together or individually. If this were the case, add the missing data.

Response 1: The shRNA molecules were applied together using a mix containing each shRNA. The missing information was added in the line 274. 

Point 2: Statistic assessments of some experiments seem not to be accurate (see below).

Figures show a limited representation of samples. I mean, only one plant per treatment (Figure 3), only one dot (Figure 4). I strongly suggest adding more pictures providing better support for the results.

Response 2: The assays were performed using three or six biological replicates for each treatment.  We just choose the most representative pictures (Figure 4)  of each treatment to ilustrate the final results.

Point 3: Some results are not described, e.g. ImmunoStrips® test shown in figure 3.

Response 3: We just mention the immunostrip test results as a symptoms confirmation test. We described just the positive results at line 106. We gave relevance to ELISA and qPCR tests as quantitative results. However, ImmunoStrip® CMV-specific kit was used following the manufacturer's protocol (Agdia Inc., Elkhart, IN, USA) mentioned in material and methods section  at line 302 of the manuscript.

Point 4: Section 2.2. Primer efficiency and specificity. It is not clear the aim of this section. I believe that Table 2 (which should be Table 1) describes the efficiency of primers used in the qPCR assays, doesn’t it? In that case, this information could be shown as a Supplementary Table.

Response 4: The reviewer is right, the table 2 is really table 1. Table 1 describes the  primers efficiency tests  (PET) used in the qPCR assays. it was changed in the manuscript.

We prefer to leave the information of PET in the main text.

Other aspects to be improved:

Line 25: Bromoviridae in italics…….It was changed in the manuscript.

Line 25: “… , widely spread worldwide” Are you qualifying the genus, the family, CMV, or everything? Clarify it…… It refers to CMV and it was corrected in the manuscript.

Line 34: Move this sentence to the end of the former paragraph……..We accept the suggestion. It was changed in the manuscript.

Line 56: “shRNAs” short hairpin RNA? Introduce it…… It was accepted and changed in the manuscript (line 158).

Line 74:  against cucumber mosaic virus. Don’t use a capital letter in this case…… it was accepted and corrected in the manuscript (line 76).

Line 80:  “The ORF2b translated respectively into viral suppressor of RNA silencing was much better expressed” ??? How many replicates?.......The manuscript was corrected, changing the word expressed by synthetized (line 83). This figure shows the electrophoretic resolution of each shRNAs after its synthesis in vitro.

RNAse reaction conditions???......Thanks to the reviewer.  The information was included in the M & M section (line 262) of the manuscript.

Line 94: Table 1 or 2???....Thanks to the reviewer. It is table 1.

Figure 2 E1 and E2. Indicate which of the shRNA was used in these assays. ……It is shRNA-1 for both treatments (DNAase/RNase).  It was added in the manuscript.

Figure 4: Could you show a larger number of samples per treatment? Besides, please include dots corresponding with non-inoculated plants treated with the shRNA solution and healthy-non-treated plants…..Thanks to the reviewer.  We just choose the most representative picture of the treatment to ilustrate the final result.

Line 99: “This protocol resulted in 100%  negative control and shRNAs-treated plants being free of symptoms and appearing resistant to CMV after 15 dpi (Figure 3a). Mild chlorosis was observed in shRNA-treated plants between 15 and 25 dpi (Figure 3c).” These two phrases convey potentially contradictory information.  After 15 dpi and between 15 and 25 dpi. What was the ratio of shRNA-treated plants with mild chlorosis after 15 dpi?........Thanks to the reviewer. The suggestion was accepted and corrected in the manuscript (lines 102).

Line 102: “Overall, symptoms detected on leaves after infiltration and inoculation treatments were consistent with ImmunoStrip® results (Figure 3a-c).” What are the results with ImmunoStrip” Describe the results in the text……It was corrected in the legend of figure 3.

Line 122: “ELISA results indicated that at…”

Line 125: Remove “However”……It is was accepted and changed in the manuscript (line 127).

Line 125: “ELISA immunoassay was able to detect 0.35 OD/mg of protein…..……It was changed in the manuscript as reviewer suggested (line 129).

Figure 5: Legend. “Different letters represent significant differences at p-value < 0.05.” Letters were not included. By the way, the bars indicating the standard errors seem to be of the same length. It is correct???? Please, verify the original data……..Thanks to the reviewer. The letters were incorporated to the figure 5 in the manuscript.

Line 136: RT- PCR and RT-qPCR….The reviewer was right. It was changed in the section 2.5 of the manuscript.

Results: The headings of this section seem to describe an M&M section e.g. “2.1 Synthesis, quality, and yield of shRNAs, 2.5. PCR and qPCR. Please, select phrases that briefly describe the results obtained in each subsection to be used as subsection headings……Thanks for the suggestion, however we prefer to leave it as it is.

Line 144. “a lower amount of PCR product when using cDNA from silenced leaves.” Where these results were shown? Besides, I think that plant leaves were not silenced. Please write this phrase.  …….This phrase was rewritten in the manuscript (line 149) according to the reviewer´s suggestion.

Figure 6. As previously observed in Figure 5, bars of standard errors in the treatments represented in figures 6 b and c seem to be of the same length. It can happen, but indeed, it is very uncommon. Please, verify the original data and the derived calculation…..The letters were unintentionality erased when transferring information to the template.  The letters were incorporated to the figure 6.

  • Legend: “a. Gene 1a. b. Gene 2a. c. gene 2b and d. gene 3b.” G or g, I think G rather than g. First in the phrase……it was corrected with G in all cases in the legend.
  • Could you add letters as a result of the statistical analysis?....yes, the letters were incorporated in the figure 6.

Line 186: Progress on the implementation (manipulation, harnessing, etc.) post-transcriptional gene silencing mechanism …Thanks, the suggestion was incorporated in the manuscript (line 174).

Line 238: “….at the 5' end. (Table 1).” Remove dot after end. I think there is something wrong with the numbering of tables. This should be Table 2……you are right…it was corrected in the manuscript (line 267).

Line 262: “Nicotiana benthamiana plants were”….Thanks, it was changed in the manuscript (line 274).

Line 285: “The Pierce ™ BCA Protein Assay Kit (Cat. 23227, 284 Thermo Scientific, Rockford, USA) was used for CP quantitation.” I can not understand how this assay specifically detected CP, rather than total protein from the plants. Explain.….Thanks to the reviewer. It was changed and explained in the manuscript (line 298).

Reviewer 2 Report

Dear Authors,

I had a great opportunity and honor to review manuscript: “Foliar infiltration of virus-derived small hairpin RNAs triggers the RNAi mechanism against the Cucumber mosaic virus” which is considered  for publication in IJMS. The CMV is member of Bromoviridae family of plant viruses. CMV ability of multi host infection (wide range of host plants) indicates it’s impact for plant production and cultivation process. Therefore, investigations of RNAi response is crucial for understanding host reaction to CMV. This manuscript presents new and interesting point of view into CMV and RNAi matter. The article is good written and show interesting and good results I have only minor critical comments to the article which is mainly connected with graphical presentation of results:

  1. Figure 2 I suggest to make the figure little smaller because gels with parts E1 and E2 has very fuzzy (many bad pixels especially on part of marker)
  2. Figure 3 I suggest to separate part d of figure as new figure . Now the part of gel is cut. I suggest to separate Figure to two one with photo of plants and one with gel
  3. Figure 4 I recommend to move into supplementary files and delete from main part of manuscript
  4. Figure 5- remove black frames on charts. Charts should be move/align into middle. Mark statistical significance values or differences on charts bards. Now the charts look like as the statistic was not performed
  5. Figure 6- remove black frames on charts. Charts should be move/align into middle. Mark statistical significance values or differences on charts bards. Now the charts look like as the statistic was not performed
  6. Table 1 move into supplementary
  7. Figure S1 Supplementary files according IJMS publication rules must be add as separate files. SO the supplementary data should not be present in main manuscript text. Supplementary data are only cited on main manuscript. Please remove all supplements from main manuscript text

Sincerely,

Author Response

Response to Reviewer 2 Comments

Main points:

  1. Figure 2 I suggest to make the figure little smaller because gels with parts E1 and E2 has very fuzzy (many bad pixels especially on part of marker)

Response 1: your recommendation is valid, but the journal instructions indicate that it is necessary to provide original photos of gels and blots. 

  1. Figure 3 I suggest to separate part d of figure as new figure . Now the part of gel is cut. I suggest to separate Figure to two one with photo of plants and one with gel

Response 2: Thanks for the suggestion. we prefer ot leave this figure as it is.

  1. Figure 4 I recommend to move into supplementary files and delete from main part of manuscript

Response 3: Thanks for your recommendation. However, we prefer leave this figure as part of the main manuscript.  

  1. Figure 5- remove black frames on charts. Charts should be move/align into middle. Mark statistical significance values or differences on charts bards. Now the charts look like as the statistic was not performed

Response 4: The frames has been removed. The letters indicating the statistical differences were included as you recommended.

  1. Figure 6- remove black frames on charts. Charts should be move/align into middle. Mark statistical significance values or differences on charts bards. Now the charts look like as the statistic was not performed

Response 5: The frames has been removed. The letters indicating the statistical differences were included as you recommended.

  1. Table 1 move into supplementary

Response 6: Thanks for your recommendation. We prefer to leave it has it is, because the primers information is important as part of the main manuscript.

  1. Figure S1 Supplementary files according IJMS publication rules must be add as separate files. SO the supplementary data should not be present in main manuscript text. Supplementary data are only cited on main manuscript. Please remove all supplements from main manuscript text

Response 7: Thank you for your recommendation. We made it following the journal’s instructions.